# Dual-Color Tracer Segmentation with Mean-Teacher Learning and Bleed-Through Correction

**Peerawat Pannattee**[1]                    PEERAWAT.PANNATTEE@RIKEN.JP
**Akiya Watakabe**[2]                           AKIYA.WATAKABE@RIKEN.JP
**Tomomi Shimogori**[2]                    TOMOMI.SHIMOGORI@RIKEN.JP
**Henrik Skibbe**[1,3]                              HENRIK.SKIBBE@RIKEN.JP

[1] *Brain Image Analysis Unit, RIKEN Center for Brain Science, Wako City, Japan*

[2] *Laboratory for Molecular Mechanisms of Brain Development, RIKEN Center for Brain Science, Wako City, Japan*

[3] *Department of Informatics, Faculty of Informatics, Matsuyama University, Matsuyama, Ehime, Japan*

## Abstract

Dual-color tracer imaging enables simultaneous visualization of multiple brain pathways, reducing experimental time. However, segmentation for quantitative signal estimation remains challenging due to the lack of ground-truth annotations, domain shift, and signal bleed-through. We propose a preliminary pipeline that combines a U-Net segmentation model trained with a mean-teacher framework and domain-specific augmentations, followed by a clustering-based method for bleed-through correction. Experimental results show improved segmentation performance and reasonable separation of tracer signals under these challenging conditions. These findings demonstrate the promise of the proposed approach for analyzing dual-color tracer data.

**Keywords:** Dual-color tracer imaging, Segmentation, Bleed-through correction, Signal unmixing

## 1. Introduction

Tracer imaging is an important technique for studying brain connectivity. It involves injecting fluorescent neural tracers into specific brain regions, where they are transported along axonal pathways. Upon imaging, the tracers emit fluorescence signals, enabling the visualization of neuronal projections and connections between different brain regions. Segmentation is then usually applied to outline the boundaries of tracer signals, which is necessary for quantitative signal estimation.

We employ dual-color (red/green) tracer injections in the marmoset brain to visualize two pathways simultaneously. This is beneficial for reducing the number of animal subjects and reduce experimental time. However, performing segmentation for the dual-color data is challenging, as it lacks reliable ground-truth annotations for training segmentation models. Although we have labeled data from previous single-color injection experiments (green only) (Skibbe et al., 2023), issues such as domain shift and signal bleed-through between color channels prevent these models from being directly applied.

This paper introduces a preliminary pipeline for segmentation and bleed-through correction in dual-color tracer data. The pipeline consists of two main steps. First, we perform

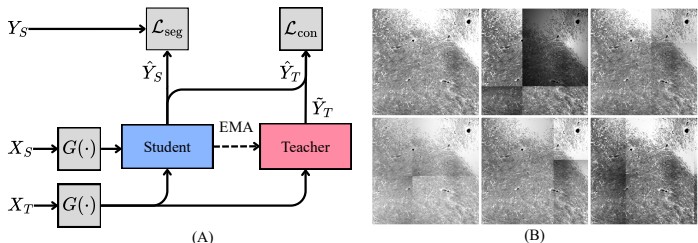

Figure 1: (A) Mean-teacher framework. (B) Examples of augmented inputs generated by $G(\cdot)$, where the top-left image is the original input.

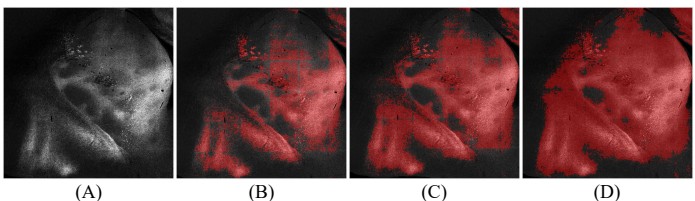

Figure 2: Preliminary segmentation results on unlabeled dual-color data (red channel). (A) Original image. (B) Supervised learning trained on single-color data. (C) Supervised learning with the designed augmentations. (D) Proposed mean-teacher framework.

segmentation on each color channel using a U-Net model trained with the mean-teacher framework, where domain-specific augmentations are designed to reduce domain shift. Second, we apply a clustering-based method to separate true tracer signals from bleed-through effects. This process produces a segmentation mask in which red and green tracer signals are separated, along with an unmixed signal image.

## 2. Method and Preliminary Results

### 2.1. Segmentation

We train a U-Net-based segmentation model (Ronneberger et al., 2015) using a self-ensemble approach based on the mean-teacher framework (Perone et al., 2019), with labeled source patches $(\boldsymbol{X_S}, \boldsymbol{Y_S})$ and unlabeled target patches $\boldsymbol{X_T}$. The model is trained with a supervised segmentation loss on the source domain (green tracer) and a consistency loss on the target domain (red/green tracer), where predictions are enforced to be consistent under stochastic augmentations, denoted as $G(\cdot)$ (Figure 1-A). We adopt two augmentations for $G(\cdot)$: grid vignetting artifacts and gamma transformation. These augmentations are designed to reflect variations observed in the target data. Grid-like vignetting artifacts frequently appear in the red channel, as residual artifacts from the image stitching process. In addition, gamma transformation is applied with a small magnitude to introduce additional nonlinear intensity variation. Figure 1-B shows examples of augmented inputs.

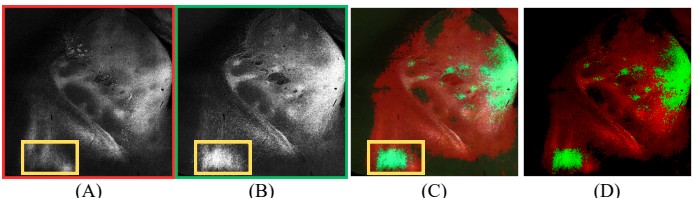

Figure 3: Bleed-through correction and unmixing results. (A) Red channel. (B) Green channel. (C) Correction result. (D) Unmixing result. Yellow boxes highlight a region that is bright in green and faint in red, correctly identified as a green tracer.

Figure 2 shows the preliminary segmentation results on unlabeled dual-color data (red channel). The baseline (B) contains noticeable artifacts and incomplete segmentation. Applying our designed augmentations can significantly reduce these issues (C). The proposed mean-teacher framework (D) further produces better segmentation. The results for the green channel are not shown, as they show little difference due to the source domain data having a similar distribution to the green channel in the target domain.

## 2.2. Bleed-Through Correction

We adopt a bleed-through correction algorithm from LUMoS (McRae et al., 2019). The key idea of this method is that each tracer produces a unique intensity distribution across color channels. The method applies a clustering algorithm to group pixels into $N + 1$ clusters, which is the number of tracers plus a background cluster. The output is constructed by assigning each pixel to a single channel based on its cluster. In our data, the background is brain tissue, which has a more complex intensity distribution than the nearly black background in LUMoS, leading to inaccurate clustering. To address this, we restrict clustering to pixels within the segmentation masks from the previous stage, assuming that all pixels belong to tracer signals. As a result, the number of clusters in our pipeline is set equal to the number of tracers.

Figure 3 shows the preliminary results of bleed-through correction. Overall, the results appear reasonable. For example, the yellow box highlights an area that is bright in the green channel and faint in the red channel, suggesting a green tracer signal. The results show that the algorithm can reasonably separate such regions.

## 3. Future Work

The current method produces promising results for segmenting and correcting bleed-through signals in dual-color tracer data. However, issues such as grid vignetting artifacts and incomplete segmentation, caused by domain shift are not fully addressed. Additionally, the current bleed-through correction algorithm performs a binary assignment, meaning that each pixel is assigned to either red or green. In practice, however, a pixel may contain a mixture of both signals. These limitations require further investigation and improvement in future work.

## Acknowledgments

This work was funded by the Multidisciplinary Frontier Brain and Neuroscience Discoveries (Brain/MINDS 2.0) from the Japan Agency for Medical Research and Development AMED JP23wm0625001.

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

## Appendix A. Implementation Details

### A.1. Input Data and Network Architecture for Segmentation

The input patches, $X_S$ and $X_T$, are randomly cropped from their corresponding full-resolution microscopy images. Each patch is first cropped to a size of $1024 \times 1024$ and then resized to $512 \times 512$ to reduce computational cost. In total, 43,520 patches are generated for $X_S$ and 13,792 for $X_T$.

We adopt a U-Net architecture, following the original design in terms of feature channel configuration, while incorporating common modern modifications, including batch normalization after each convolution and bilinear upsampling with an additional $1 \times 1$ convolution in the decoder.

### A.2. Training Objective for the Mean-Teacher Framework

The training objective consists of a supervised segmentation loss and an unsupervised consistency loss. The supervised loss is applied to labeled source data $(X_S, Y_S)$ and is defined as the pixel-wise cross-entropy loss:

$$\hat{Y}_S = f(X_S), \tag{1}$$

$$\mathcal{L}_{\text{seg}} = -\sum_i Y_S^{(i)} \log\left(\hat{Y}_S^{(i)}\right), \tag{2}$$

where $f(\cdot)$ denotes the segmentation model and $i$ indexes spatial locations.

For unlabeled target data $\boldsymbol{X_T}$, a consistency loss is applied between the student and teacher model predictions under stochastic augmentations. Let $G_1(\cdot)$ and $G_2(\cdot)$ denote two independently augmented versions of the input generated by the augmentation function $G(\cdot)$. The consistency loss is defined as:

$$\hat{\boldsymbol{Y}}_{\boldsymbol{T}} = f_s\big(G_1(\boldsymbol{X_T})\big), \tag{3}$$

$$\tilde{\boldsymbol{Y}}_{\boldsymbol{T}} = f_t\big(G_2(\boldsymbol{X_T})\big), \tag{4}$$

$$\mathcal{L}_{\mathrm{con}} = \left\|\hat{\boldsymbol{Y}}_{\boldsymbol{T}} - \tilde{\boldsymbol{Y}}_{\boldsymbol{T}}\right\|_2^2, \tag{5}$$

where $f_s$ and $f_t$ denote the student and teacher models, respectively.

The overall training objective is given by:

$$\mathcal{L} = \mathcal{L}_{\mathrm{seg}} + \mathcal{L}_{\mathrm{con}}, \tag{6}$$

The teacher model parameters are updated as an exponential moving average (EMA) of the student parameters.

### A.3. Input for the Bleed-through Correction Algorithm

The bleed-through correction algorithm operates on full-resolution microscopy images without patching. Each image is first resized such that its longest side is at most 4048 pixels. The image is then flattened, where each pixel is treated as an individual input sample represented by its three-channel intensity values. The clustering process is performed independently for each image.

## Appendix B. Supplementary Results

Figure 4 shows qualitative segmentation results on unlabeled dual-color data. The proposed mean-teacher framework produces more consistent and accurate predictions compared to supervised baselines.

Figure 5 presents the bleed-through correction results. The method effectively separates true tracer signals from cross-channel interference.

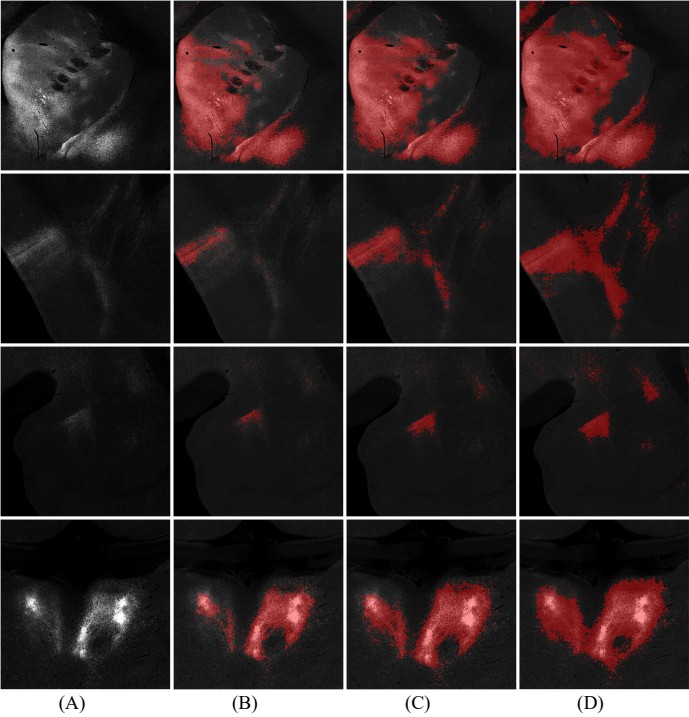

(A)  (B)  (C)  (D)

Figure 4: Segmentation results on unlabeled dual-color data (red channel). (A) Original image. (B) Supervised learning trained on single-color data. (C) Supervised learning with the designed augmentations. (D) Proposed mean-teacher frame-work.

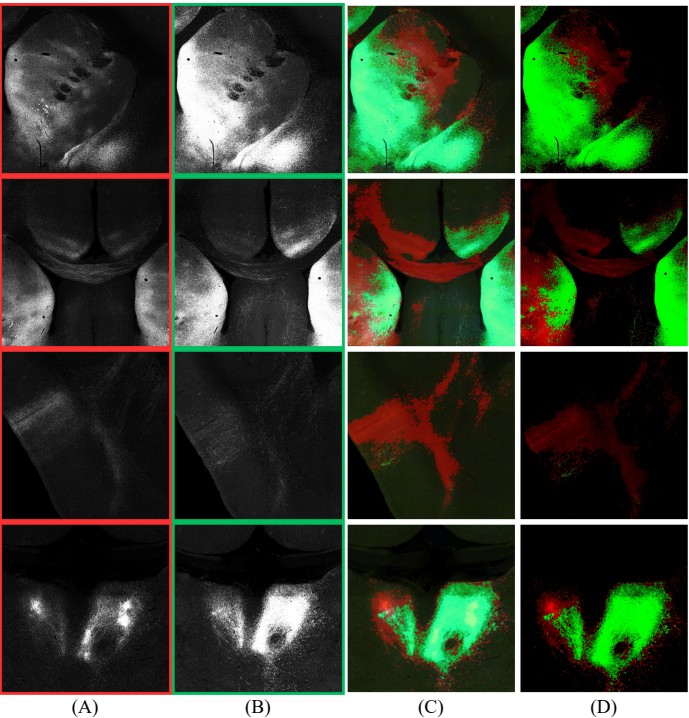

(A)    (B)    (C)    (D)

Figure 5: Bleed-through correction and unmixing results. (A) Red channel. (B) Green channel. (C) Correction result. (D) Unmixing result.

