# OpenReview forum: "Dual-Color Tracer Segmentation with Mean-Teacher Learning and Bleed-Through Correction"
_MIDL.io/2026/Short_Papers — MIDL 2026 - Short Papers Poster_

### Official Review · Reviewer_M6Xs · 2026-04-24
**A Specialized Pipeline with Limited Validation**

**Rating:** 3
**Confidence:** 2

**Review:**

See strengths and weaknesses.

**Summary:**

The paper presents a pipeline for segmenting dual-color tracer images and separating overlapping signal traces. The approach consists of two primary components. First, a mean teacher framework is employed to enable semi-supervised learning from unlabeled image patches. Second, the authors introduce an enhanced bleed-through correction module that extends a previously published method to improve signal separation.

**Strengths:**

The use of a mean teacher model is a sensible choice for leveraging unlabeled data in a semi-supervised setting. The overall pipeline is logically structured, and the preliminary qualitative results suggest that the method can produce reasonable segmentation outputs.

**Weaknesses:**

Both the dataset and the proposed pipeline appear to be highly specialized to a particular imaging modality and experimental setup, which may limit broader applicability. Additionally, the paper lacks quantitative evaluation.

**Justification Of Rating:**

the contribution appears narrowly scoped to a particular application. Combined with the limited empirical validation, this constrains the overall impact. Although the methodological choices are appropriate, the level of novelty and general significance may not be sufficient to support a strong accept. The reviewer has limited familiarity with the specific imaging modality.

---

### Decision · Program_Chairs · 2026-05-08

Accept (Poster)